# SSDBN: A Single-Side Dual-Branch Network with Encoder–Decoder for Building Extraction

Yang Li [1], Hui Lu [1], Qi Liu [1,*], Yonghong Zhang [2] and Xiaodong Liu [3]

1    School of Computer and Software, Engineering Research Center of Digital Forensics, Ministry of Education, Nanjing University of Information Science & Technology, Nanjing 210044, China; 20201220020@nuist.edu.cn (Y.L.); 20191221006@nuist.edu.cn (H.L.)
2    School of Automation, Nanjing University of Information Science & Technology, Nanjing 210044, China; zyh@nuist.edu.cn
3    School of Computing, Edinburgh Napier University, Edinburgh EH10 5DT, UK; x.liu@napier.ac.uk
*    Correspondence: qi.liu@nuist.edu.cn

**Abstract:** In the field of building detection research, an accurate, state-of-the-art semantic segmentation model must be constructed to classify each pixel of the image, which has an important reference value for the statistical work of a building area. Recent research efforts have been devoted to semantic segmentation using deep learning approaches, which can be further divided into two aspects. In this paper, we propose a single-side dual-branch network (SSDBN) based on an encoder–decoder structure, where an improved Res2Net model is used at the encoder stage to extract the basic feature information of prepared images while a dual-branch module is deployed at the decoder stage. An intermediate framework was designed using a new feature information fusion methods to capture more semantic information in a small area. The dual-branch decoding module contains a deconvolution branch and a feature enhancement branch, which are responsible for capturing multi-scale information and enhancing high-level semantic details, respectively. All experiments were conducted using the Massachusetts Buildings Dataset and WHU Satellite Dataset I (global cities). The proposed model showed better performance than other recent approaches, achieving an F1-score of 87.69% and an IoU of 75.83% with a low network size volume (5.11 M), internal parameters (19.8 MB), and GFLOPs (22.54), on the Massachusetts Buildings Dataset.

**Keywords:** building extraction; dual-branch; semantic segmentation; encoder–decoder network

## 1. Introduction

Image segmentation, target detection, image classification, semantic segmentation, and instance segmentation are five major areas of computer vision [1], and semantic segmentation is an effective way to conduct the pixel-level classification of images, which are assigned labels to represent the categories to which they belong after they have passed through a semantic segmentation network. Such processes can therefore be understood as a dense classification problem [2–5].

At present, the quantity and quality of modern remote sensing images are continuing to increase, making it more convenient for researchers to obtain high-resolution remote sensing images. These remote sensing images play an important role in the fields of land and resource exploration, environmental detection and protection, urban planning, crop yield estimation, and disaster prevention and mitigation [6–8]. However, how to effectively and efficiently extract relevant information from such large-scale images has become challenging. Precisely identifying ground buildings in high-resolution satellite images is necessary to conduct consequent tasks, e.g., updating urban spatial databases, detecting urban dynamic changes, and establishing smart cities [9–11].

Moreover, the extraction of surface buildings from the finished product of satellite imagery suffers from two potential problems, i.e., synonym spectrum and foreign objects

with the same spectrum [12]. Such problems happen since remote sensing information (e.g., light, shadow, texture, and color) varies over time. Buildings are similar in color to land and grass, so they can easily to be missed or wrong judged; sometimes, a building may appear to comprise different buildings due different architectural styles and/or texture characteristics. It is the diversity of these factors that affects extraction accuracy, but it is tough to find a perfect solution for all issues.

Recent extraction efforts of surface buildings can be divided into two categories: traditional approaches using unsupervised learning algorithms and data-driven learning methods based on deep learning models.

The traditional unsupervised methods mainly use the characteristics of various types of buildings in remote sensing images, such as spectrum, geometry, and the correlation between pixels, to extract buildings [13,14]. Feature extraction is performed with designed algorithms, and statistical learning models such as support vector machines [15], decision trees [16], and clustering [17] are used for feature classification. Generally speaking, feature extraction and classification models are separated and can be freely combined, but their accuracy and efficiency are not high enough.

A deep learning network uses the cross-epoch improvement method. First, the machine autonomously learns the relationship between potential pixels and categories with a large amount of data, which greatly improves the accuracy and speed of classification. Researchers do not need to design algorithms to perform feature extraction and realize the semi-black box mode. Due to the down- and up-sampling of convolutional neural networks (CNNs), an end-to-end framework can be realized.

Since the fully convolutional neural network was proposed in 2015, it has been widely used in various semantic research fields [18–20]. A series of improved methods based on complete convolution networks (FCNs) have been proposed [21–25]. Although these proposed methods have achieved good results in the extraction of buildings from remote sensing images, they also have some problems:

- High-resolution remote sensing images can clearly express the spatial structure of ground objects and contain texture features and details, include the potentially changing tones and characteristics of some individual buildings. The incomplete or incorrect extraction of semantic information may occur.
- To accurately extract more features, some researchers have adopted the network scale method. By deepening the network and increasing the number of network calculations, however, network parameters can comprise hundreds of megabytes, which consumes memory and leads to slow loading speeds during real-time forecasting.

To better address the issue of low extraction accuracy in high-resolution photos caused by intricate details, the authors of this paper propose a new decoder network called a dual-branch decoder, in which the decoder network transmits the features extracted by the central module to the deconvolution and feature enhancement branches of the decoder, which captures multi-scale information and high-level semantic information and further improves the accuracy and detail of building extraction by correcting the residuals between the main branch of the decoder network and the feature mapping. A high-resolution image usually has tens of thousands of pixels; the background can account for up to 99% of some images, and most buildings do not comprise 40% of images. This type of problem is called data imbalance. This problem has often appeared in previous research, so we used a previously designed solution here: in the model design, a loss function called dice loss was used [24]. Setting dice loss can effectively solve the problem of data imbalance and further improve the accuracy of networks for building feature extraction.

The contributions of this paper is as follows:

1. The authors of this paper propose a lightweight dual-branch encoder–decoder framework. A comparison of multiple deep learning networks on the Massachusetts Building Dataset and WHU Satellite Dataset I (global cities) showed that the network is suitable for building extraction in remote sensing images.

2.   On the Massachusetts Building Dataset and WHU Satellite Dataset I (global cities), it was proven that in the training phase, adopting dice loss as the loss function was more favorable to easing data imbalance in the building extraction task than binary cross entropy (BCE) loss. A combination of the Xception and Res2Net networks was able to effectively replace Visual Geometry Group Network 16(vgg16) and greatly reduce the number of network parameters.

3.   The authors of this paper propose a new dual-branch module that can ensure a network retains suitable accuracy in deeper situations and prove that the dual-branch decoder could be easily migrated to other networks.

The remainder of this paper is structured as follows. Section 2 summarizes the related work of semantic segmentation, context information, and architectural segmentation extraction. In Section 3, we describe our SSDBN method in detail. Detailed discussions of experiments and results are presented in Section 4. Finally, the conclusions of this paper are shown in Section 5.

## 2. Related Work

### 2.1. Semantic Segmentation

Most of the work of semantic segmentation is realized and advanced with deep networks. In the past few years, computing power has rapidly expanded and developed, and semantic segmentation networks based on deep learning have experienced great advancements. Since the FCN was proposed in 2015 [25], the use of fully convolutional networks to restore images to achieve significant end-to-end effects has been a focus of the field. Generally, through fine adjustments to FCNs, end-to-end networks can be more adapted to the requirements and challenges of the new era. In 2017, Chen et al. [26] proposed DeepLab by introducing conditional random fields (CRFs) and dilated convolution refinement results. To reduce the number of the parameter in the original FCN, Badrinarayanan et al. [27] optimized the pool layer parameters of the decoder. The proposed SegNet uses un-pooling instead of un-sampling so that the number of parameters of the whole network are optimized to half that of the FCN. In contrast to SegNet, DeconvNet [28] uses a fully connected (FC) layer in the middle as a term to strengthen category classification.

With today's increasingly complex datasets, the drawbacks of FCNs have been exposed: the receptive field is too small to obtain global information and it is difficult to capture spatial correlations, so results are fuzzy. To fix these issues, UNet was proposed [29]. UNet supplements the decoder network by combining features from the encoder network, thus allowing the entire network to deepen the number of network layers while supplementing the details of the original features to obtain more accurate outputs.

### 2.2. Context Information

Context information aggregation is a common method used to increase the pixel representation in a semantic segmentation network. The original FCN does not explicitly use global context information. In order to address this issue, ParseNet of ICLR2016 [30] uses global pooling to calculate global features to enhance the feature receptive field of each pixel, which can effectively obtain very rich context information. CVPR2017's PSPNet [31] uses a pyramid pooling module to obtain multi-scale contextual information so that the problems of different pictures, different scales, and difficult-to-identify objects are solved. Deeplabv2, v3 [32], mainly adjusts pyramid pooling using dilated convolution instead of general convolution operations, and it can extract rich multi-scale context information. The OCR [33] method traces its roots to and is consistent with the original definition of the semantic segmentation problem; it mainly converts the pixel classification problem into the object region classification problem so that it can explicitly enhance the global information of an object. The GMEDN [34] model captures multi-scale contextual information with a knowledge distillation module.

### 2.3. Building Segmentation

Pan et al. [35] presented a spatial and channel attention mechanism (SCA)-based generative countermeasure network for accurate building segmentation. Protopapadakis et al. [36] proposed a deep neural network (DNN) based on stack automatic encoder (SAE) drive and semi-supervised learning (SSL) to extract buildings from low-cost satellite imagery. Wang et al. [37] proposed a new type of nonlocal residual U-shaped network that uses the codec structure to extract and recover feature maps and that obtains global context information using the self-attention method. Hu et al. [38] built new modules by setting up their components to improve progress, and an attention mechanism was added to the network to improve segmentation accuracy. Liu et al. [39] proposed a network that could recover the details of a lightweight model through a spatial pyramid. An adaptive iterative segmentation method was presented by Chen et al. [40]. Cheng et al. [41] presented a DARNet (deep active ray network) for end-to-end training through the energy minimization and backpropagation of a backbone CNN to obtain accurate building segmentation. Shi et al. [42] integrated a graph convolution network (GCN) and deep structure feature embedding (DSFE), and they proposed a gated graph convolution network to generate clear boundaries and fine-grained pixel-level classification.

As mentioned above, although a lot of work has been done in the field of building extraction by applying semantic segmentation and context knowledge in the visual field, there is still much room for accuracy improvements in practical application of building extraction under different conditions, and most depth networks have complex parameters and large time and space complexity that affect the accuracy and efficiency of building extraction. Accordingly, the method proposed in this paper uses the semantic segmentation of high-resolution remote sensing images with a light weight model structure that captures deeper semantic information, improves the accuracy of building extraction, and reduces the complexity of the model structure.

## 3. Methodology

### 3.1. Overall Architecture

Figure 1 shows the general framework of the building network proposed in this paper. It can be divided into three parts: a lightweight backbone network encoder, a multi-scale middle layer component, and a dual-branch decoder. In the encoder module, the Xception network is used as the backbone network [43]. This is because the Xception backbone network is necessary when adding residual connections, which can effectively enhance the transmission of information and help address the problems of overfitting and gradient disappearance caused by deep learning networks. In addition, the feature information extracted by the Xception backbone network is conducive to obtaining the relevance and spectral characteristics of a remote sensing network space. Based on the above considerations, the backbone network used in this paper was Xception, and the improved Res2Net was used in the backbone network instead of ordinary convolution to address the problem of deep network degradation. Following the backbone network is the multi-scale middle layer component, which is divided into two parts: the attention and feature fusion modules. The feature map extracted by the attention module and then processed by the feature fusion module can effectively obtain rich multi-scale surface building information. In the decoder module, we constructed a two-branch decoder architecture, which included a deconvolution branch and a feature enhancement branch. The deconvolution branch is responsible for capturing basic information and adding low-level semantic details, and the feature enhancement branch is used to further strengthen high-level semantic information and deepen multi-scale information. This is an important design feature for this network model that can be used to achieve superior results.

The details of Res2Netplus are discussed in Section 3.2. The details of convolutional block attention module (CBAM) are discussed in Section 3.3. The feature fusion module can fuse higher semantic information, and its details are discussed in Section 3.4. The

two-branch network is divided into a deconvolution branch and a feature enhancement branch, the details of which are discussed in Section 3.5.

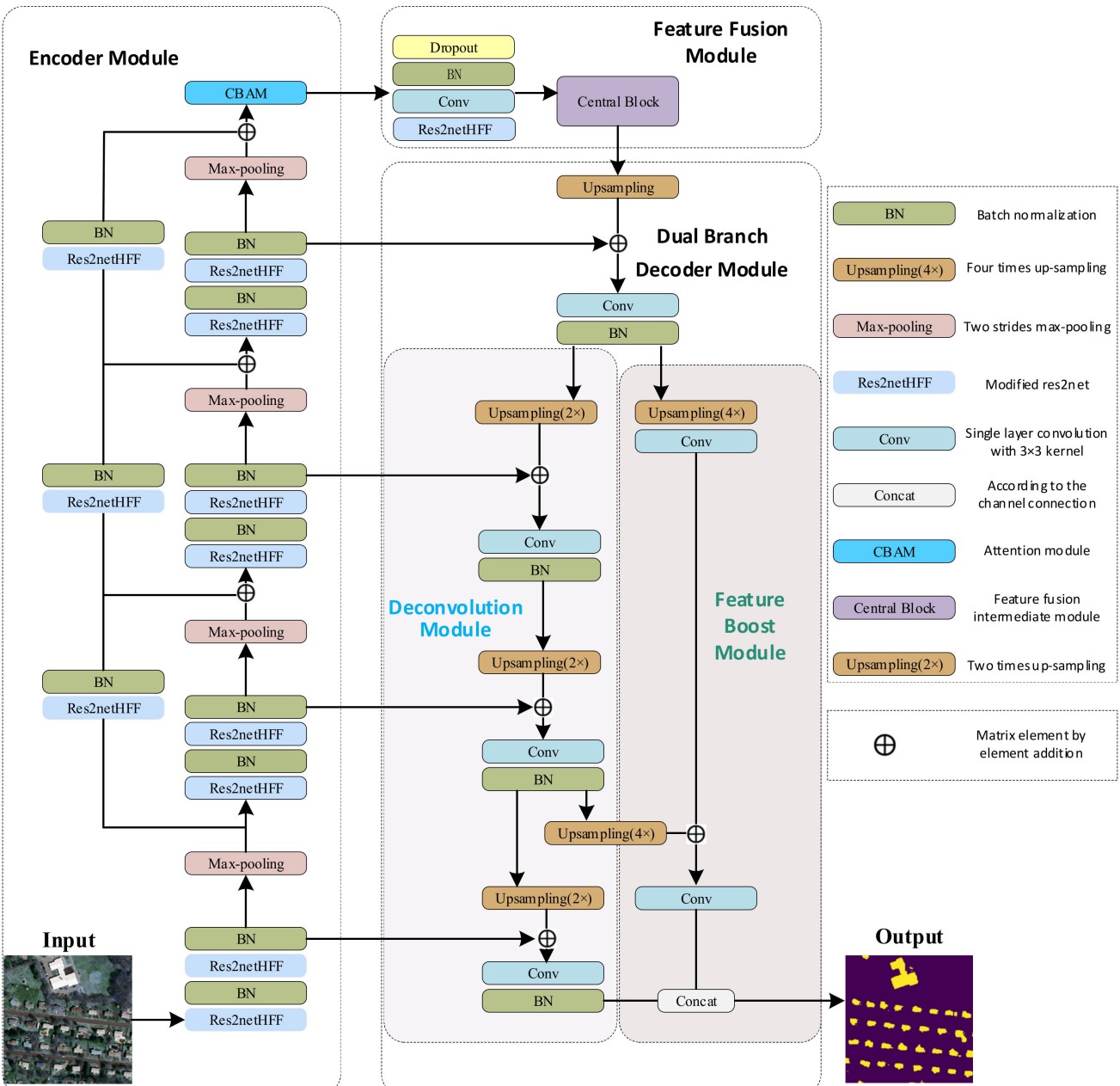

**Figure 1.** Overall Architecture of our single-side dual-branch network (SSDBN).

### 3.2. Res2Netplus

In semantic segmentation, a large number of backbone networks in ordinary convolutional networks realize the most advanced information. The main realization method is to extract richer semantic information and wider receptive fields. These measures are often taken to increase network depth. However, with the deepening of the network level, the performance of the network gradually increases to the saturated state and then rapidly decreases, that is, the problem of network degradation occurs and it becomes difficult to obtain good training results. On this basis, ResNet was proposed [44], and it has been found to solve this problem well. However, as the network level deepens, the network parameters also increase, thus increasing the model's training time and memory consumption; along with relatively high computational and time costs, these issues make it difficult to enact

real-world applications. In the previously proposed MobileNet architecture [45], the core idea to reduce the weight of the network model is to use deep separable convolution to replace the ordinary convolutional layer, thus reducing the model parameters, but the problem is that the model's training accuracy is reduced. Accordingly, to solve the problems of network degradation, increased parameter amounts, and a lack of multi-scale information at deeper network levels, the authors of this paper adopted a new Res2Net structure [46]. This convolution method reduces the number of network parameters without increasing the computational load, and it contains rich multi-scale information.

According to Figure 2a,b, compared to the general convolution method, Res2Net mainly replaces a set of 3×3 convolutions with multiple sets of smaller 3×3 filters, with hierarchical residuals presenting different ways to connect different filter groups.

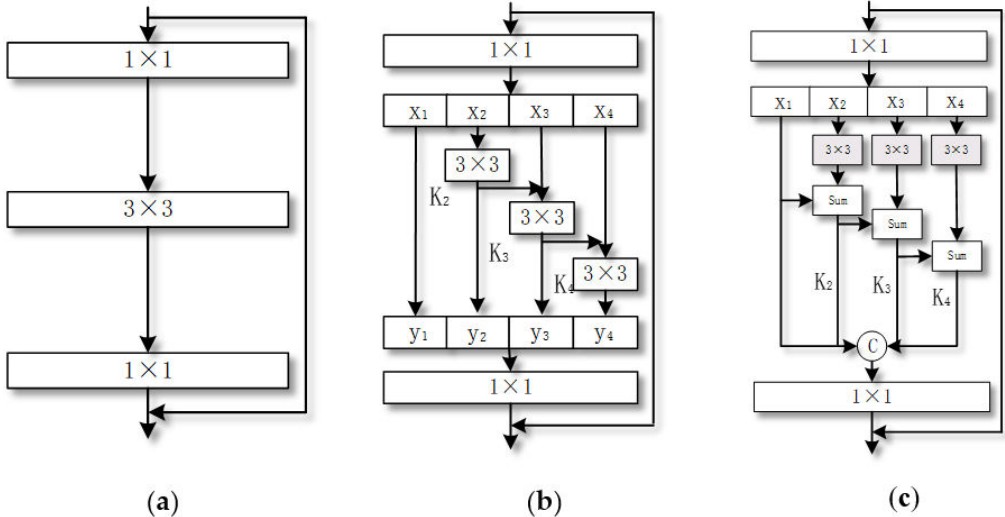

**Figure 2.** Improved Res2Net module. (**a**). ResNet [43]; (**b**). Res2Net [45]; (**c**). Res2Netplus.

It is worth mentioning that the authors of this paper made use of the method of inputting equal interval sampling in the center of 3×3 convolution kernels to replace ordinary convolution with dilated convolution, mainly to expand the receptive field without losing the resolution of remote sensing image. Moreover, dilated convolution has a parameter that can set the dilation rate, arbitrarily expand the receptive fields, and capture multi-scale context information at the same time. In short, the introduction of dilated convolution can arbitrarily expand the receptive field without introducing additional parameters, which makes the model more light weight.

First, the high-dimensional feature map is projected into the low-dimensional space through a 1×1 convolution kernel. Then, the obtained feature map F is equally divided into four parts according to the channels, and the output features of the previous group and the input features of the next group are sent to the next 3×3 filter. This procedure is continued until all four feature portions have been processed. Then, all the features are combined, the 1×1 convolution kernel is used to adjust the channel, and the information is fused. Compared with the original set of filters, the perceptual field is increased and many equivalent feature scales are generated due to the combination of different filters. In previous paper on Res2Net [46], it was suggested that it is not necessary to divide the map into four groups; the number can be changed according to the requirements of different networks. In this experiment, we divided the characteristic map into four groups according to channels, and we conducted ablation experiments (discussed in Section 4.5.2) to prove that our changes were effective. The improved formula is as follows:

$$
y_i = \begin{cases}
x_i & i = 1; \\
K_i(x_i) & i = 2; \\
K_i(x_i + y_{i-1}) & 2 < i \leq s.
\end{cases} \tag{1}
$$

where $i$ represents the index of channels, $x_i$ represents the input of each channel, $K_i$ represents that each $x_i$ has a corresponding 3×3 convolution, and $y_i$ represents the output of $K_i$. Specifically: the output of the feature subset $x_i$ and $y_{i-1}$ is added together and then used as the input of $K_i$. Simultaneously, to minimize the parameters, the 3×3 convolution of $x_1$ is omitted while the repeated use of features is realized. Finally, the four outputs are fused through a 1×1 convolution, and the result is output. This method not only efficiently processes the feature information contained in an image but also obtains different numbers of different receptive field sizes, which greatly reduces the computational overhead.

However, due to the characteristics of high-resolution remote sensing images themselves), the original Res2Net module cannot extract information from high-resolution images well due to its low coupling fusion. To tackle the problem of fine-grained level and multi-scale extraction, the authors of this article propose improvements of Res2Net, the details of the which are as follows:

1. Res2Net is a stack of ordinary convolutional layers. In general, compared to dilated convolution with the same convolution kernel size, the receptive field of an ordinary convolution kernel is not open enough to capture deeper information. This paper mainly deals with the extraction of buildings in the field of remote sensing. Compared with other fields, the extracted target scales are different, so we chose dilated convolution in the Res2Netplus proposed in this article to improve the receptive field.

2. As shown in Figure 2c, as with the original Re2Net, the high-dimensionality of the feature map is projected into the low-dimensional space through a 1×1 convolution kernel. Then, the obtained feature map F is equally divided into 4 parts according to the channel. The difference from the original version is that the output features obtained after each X input are subjected 3×3 dilated convolution and sequentially added. This process is repeated several times until all four feature parts are processed. Then, all features are combined, a 1×1 ordinary convolution kernel is used to adjust the channel, and the feature information is fused. We propose this modification due to the characteristics of high-resolution remote sensing images, which contain complex information, diverse texture feature types, and variability. The detailed information contained within them is very rich, and the image have multi-scale characteristics. The information content and the level of detail reflected by different scales are also different. Therefore, when each pixel in a low-dimensional space is input, the feature is extracted through a 3×3 dilated convolution. Through the function of the dilated convolution, the receptive field can be expanded and more detailed contextual feature information of different scales of the image can be captured, so each output feature can be added to obtain deeper multi-scale information of the high-resolution image. Our improved Res2Netplus formulas are shown in Equations (2) and (3), which are used as the most basic unit of the encoder part of the network model to replace the ordinary convolution in the backbone Xception network. At the same time, in order to test the effectiveness of the improved Res2Net, we conducted an experimental demonstration.

$$K_i = \begin{cases} x_i & i = 1; \\ K_i(x_i) + K_{i-1} & 2 \leq i \leq s \ (s = 4). \end{cases} \tag{2}$$

$$C = concat(x_1, K_2, K_3, K_4) \tag{3}$$

where the concat ($\cdot$) represents the addition of the features obtained from the four groups of feature maps $x_1$, $K_2$, $K_3$, and $K_4$ after dilated convolution.

According to the evaluation indicators obtained from the ablation experiment discussed in Section 4.5.2, our method showed improvements. Additionally, Figure 3 demonstrates that our improved method was very effective. The network of Figure 3a was not improved regarding the original Re2Net processing effect, although a basic feature image could be obtained. However, Figure 3b demonstrates that the image features became clearer following the replacement of the ordinary convolution with dilated convolution. The impact shown in Figure 3c is more significant. With our improved method, we could

obtain deeper visual information by performing dilated convolutions and then sequentially fusing them. Suppose the size of an input image is $H_0 \times W_0 \times C_0$, which, respectively, represent the height, width, and number of channels of an image; the feature is X after being output by the Xception backbone network. The expression of this process is:

$$X = F_{Xception}(Input, W_{Xception}) \tag{4}$$

where $F_{Xception}(\cdot)$ represents the mapping function of the input image through the backbone network Xception and $W_{Xception}$ represents the learning weight of Xception.

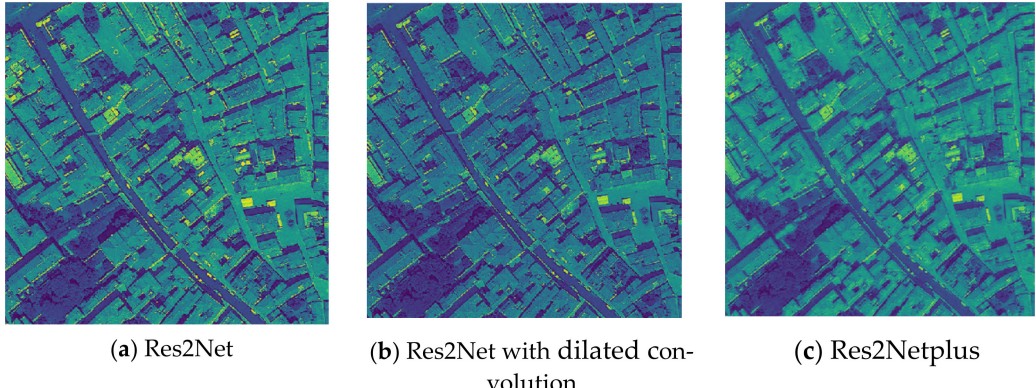

|  |  |  |
|:---:|:---:|:---:|
| (**a**) Res2Net | (**b**) Res2Net with dilated convolution | (**c**) Res2Netplus |

**Figure 3.** Comparison of improved Res2Net: (**a**) the result of Res2Net; (**b**) the result of Res2Net with dilated convolution; (**c**) the result of Res2Netplus with our proposed.

The weight can be understood as the importance of the control signal. In this paper, it represents the important pixels expressing building information in the deep network learning remote sensing image. The weight changes until the best value is learned.

*3.3. CBAM*

Extracting buildings from remote sensing images usually involves the extraction of multi-scale information. Most common public building datasets are based on residential areas, and the biggest feature of these datasets is that the buildings are continuous, with large-scale information. However, when training a network model, it is not possible to effectively extract accurate semantic information for some small-scale buildings with unclear feature information suddenly appearing, thus leading to a network model that is not robust. In effect tests, small buildings are often ignored, and the testing effect is not very satisfactory. Therefore, our previous proposal of the attention mechanism [47] has shifted focused to many model networks and the obtainment of more detailed information of chosen targets at the expense of other useless information. In this paper, the use of this attention mechanism led to good results in building extraction. In layman's terms, the attention mechanism is used so that a network can automatically learn where an image needs attention, and the authors of this paper used it to make their network learn to ignore the irrelevant information of non-buildings and pay attention to building information, especially building edge extraction. At the same time, it is hoped that the small-scale building information contained in images can be automatically learned. To achieve the multi-scale semantic information extraction of buildings, the authors of this paper mainly used the convolution block attention module (CBAM) [48]. Although this attention mechanism was proposed earlier, it is not a non-local, but there is has fewer parameters and therefore makes the model lighter and able to achieve better results.

In Figure 4, it can be seen that the CBAM attention mechanism module is mainly divided into two modules: channel and space attention. The network learns the attention points and the expression of attention points in the channel and space dimensions, and then it increases the expressiveness of the features by focusing on important features and suppressing non-essential features.

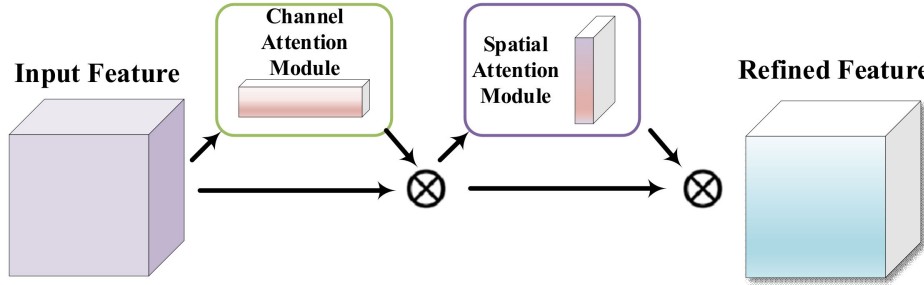

Matrix element by element multiplication

**Figure 4.** The framework of our CBAM.

In this paper, the feature map derived from the network's backbone was input to the CBAM module. In this module, the output of the feature by the backbone network is first input to the channel attention module as input features. Then, new features are obtained through channel attention. Formula (5) expresses the calculation process of channel attention.

$$
\begin{aligned}
F'_C(X) &= \sigma(MLP(AvgPool(X)) + MLP(MaxPool(X))) \\
&= \sigma(W_1\left(W_0\left(X^C_{avg}\right)\right) + W_1\left(W_0\left(X^C_{max}\right)\right)
\end{aligned}
\tag{5}
$$

where $\sigma$ represents the use of the sigmoid activation function, $W_1$ represents the attention weight, $C$ represents the channel, and MLP represents the non-linear fitting operation. Formula (6) expresses the calculation process of spatial attention.

$$
M_s(X') = \sigma\left(f^{7\times7}\left(\left[AvgPool(X'); MaxPool(X')\right]\right)\right) = \sigma(f^{7\times7}\left[X'^s_{avg}; X'^s_{max}\right]
\tag{6}
$$

where $\sigma$ represents the use of the sigmoid activation function and $f^{7\times7}$ represents the convolution operation through the 7×7 convolution kernel. The CBAM module can effectively focus the overall attention of the model on the non-background. As discussed in Section 3.2, *Atten* is used to represent the output of X after the CBAM module is used. The expression of this process is:

$$
Atten = F_{CBAM}(X, W_{CBAM})
\tag{7}
$$

where $F_{CBAM}(\cdot)$ represents the mapping function of X through CBAM and W represents the learning weight of the CBAM module.

### 3.4. Feature Fusion

The authors of this paper tried to perform a large-scale extraction of the feature information extracted from the backbone network, that is, adding feature fusion in the middle layer of the network so that more semantic information could be extracted in a small area. As shown in Figure 5, the specific method consisted of expanding the receptive field of image feature extraction; using 512 3×3 convolution kernels; setting the sampling expansion rate sequentially to 1, 2, 4, 8, 16, and 32 for the convolution operation; and convoluting. The features are batch-normalized and finally activated by the ReLU activation function. The number of decoders corresponds to the number of encoders, and each decoder includes one deconvolution layer and two convolution layers. The deconvolution layer up-samples the features obtained in the previous stage twice and connects with the encoder of the same level to restore the image size of each layer in the encoder stage while performing feature extraction through two convolutional layers. The goal of this process is to fuse the characteristic information output with the deconvolution operation of different proportions and finally restore the original input image size, thereby preserving richer information.

*Enc* is used to represent the output after feature fusion. This feature fusion process can be expressed as:

$$Enc = MLP_{fuse}\left(feature, W_{fuse}\right) \quad (8)$$

where $MLP_{fuse}(\cdot)$ represents the mapping function of *Atten* through the feature fusion module and w represents the learning weight of the feature fusion module.

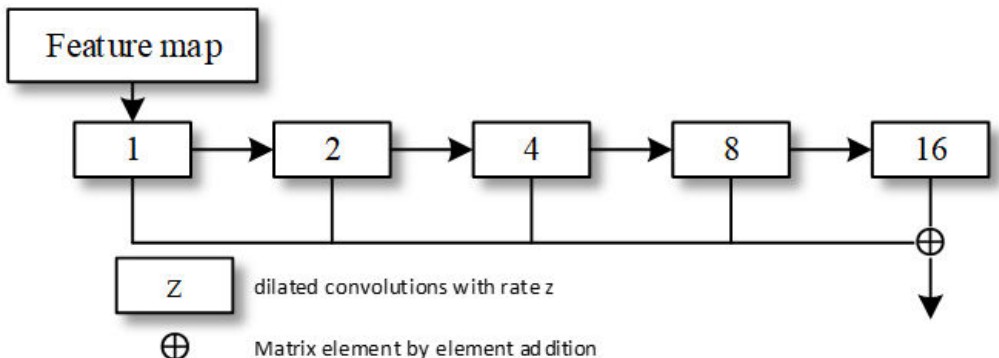

**Figure 5.** The module of feature fusion.

*3.5. Dual-Branch Decoder*

In the decoder module, we built a dual-branch decoder structure that mainly includes a deconvolution branch and a feature enhancement branch. The deconvolution branch is used to capture basic feature information and supplement potential semantic information. The feature enhancement branch enhances semantic information, deepens multi-scale information, and further refines weak semantics and coarse semantic predictions.

In the deconvolutional branch, there are four deconvolution layers. Each deconvolution layer consists of double up-sampling, a convolution operation with a convolution kernel size of $3 \times 3$, and the batch normalize (BN) operation; two-times up-sampling and convolution operations are used for the deconvolution of the decoder, and the BN operation is used to prevent the network gradient dispersion or even disappearance. This process is represented by $F'_{deconvolution}$. In Figure 1, Enc is the input of the deconvolutional branch. The deconvolution layer size is $D_i \in R^{H_{D_i} \times W_{D_i} \times C_{D_i}}$, $(i = 1, 2, 3, 4)$. This process can be described as:

$$F_{deconvolution} = Add\left(Enc, F'_{deconvolution}\right) \quad (9)$$

where the $F_{deconvolution}$ denotes the deconvolutional branch and *Add* represents the sum function of encoder block and decoder block features.

Therefore, to extract building information of different scales from high-resolution remote sensing images and refine potential semantic information, feature enhancement branches must be constructed. One branch continues the deconvolution operation set by the model and the other branch performs the feature enhancement operation. The feature enhancement branch is mainly completed by four-times up-sampling and $1 \times 1$ convolution operation. The convolution of $1 \times 1$ is used to reduce a high dimension feature map to a low dimension map. In this way, the feature map obtained in this paper is expressed as:

$$FE \in R^{H_D \times W_D \times C_D} \quad (10)$$

where $H_D \times W_D \times C_D$ denote every deconvolutional layer shape.

Then the results obtained by the feature enhancement branch are fused, the number of channels available is modified through $1 \times 1$ convolution, and the convolution is finally connected with the output of the last layer of deconvolution operation. The final result

contains both basic feature data and high-level semantic data. The outputs are presented by $F\_output \in R^{H_0 \times W_0 \times Class}$. The following formula can be used to describe this process:

$$F\_output = \text{concat}(F_{deconvolution}, FE) \tag{11}$$

where the concat (·) connects the output of deconvolution with the output of two branches, and the class is 2.

Finally, a binary picture of the same size as the input image is output to represent the building extraction result. The semantic category corresponding to the input image is obtained with the argmax function. In this paper, binary prediction was used to indicate whether each pixel belonged to a building or background.

## 4. Experiment and Result

### 4.1. Dataset

The experiment of this paper was based on the Massachusetts Buildings Dataset [49] and WHU Satellite Dataset I (global cities) [50], which are public datasets that can be directly downloaded through official websites.

The Massachusetts Buildings Dataset is a collection of 151 high-resolution aerial images of the Boston, Massachusetts area. Each high-resolution image has a resolution of 1500×1500. Each image can cover an area of 2.25 square kilometers. Thus, the whole dataset covers a total of 339.75 square kilometers in the Boston area. The data processing strategy used in this experiment was use sliding window cropping. On this dataset, we use 156 sliding windows to slide up, slide down, and crop in a horizontal and vertical manner. The cropped training set contained 8768 512×512 images. We then performed the selection for the final dataset, in the training set contained a total of 7488 256×256 images, the test set had 640 images, and the validation set had a total of 256 images.

The Massachusetts Buildings Dataset has three major advantages: a large area, a wide density of building distribution, and a wide variety of buildings. In building extraction neural networks, these advantages can provide high fit and generalization of network parameters driven by the large number and type of buildings.

Figure 6 presents two sample images from this dataset. The diversity of building styles and the density of distribution can be clearly observed.

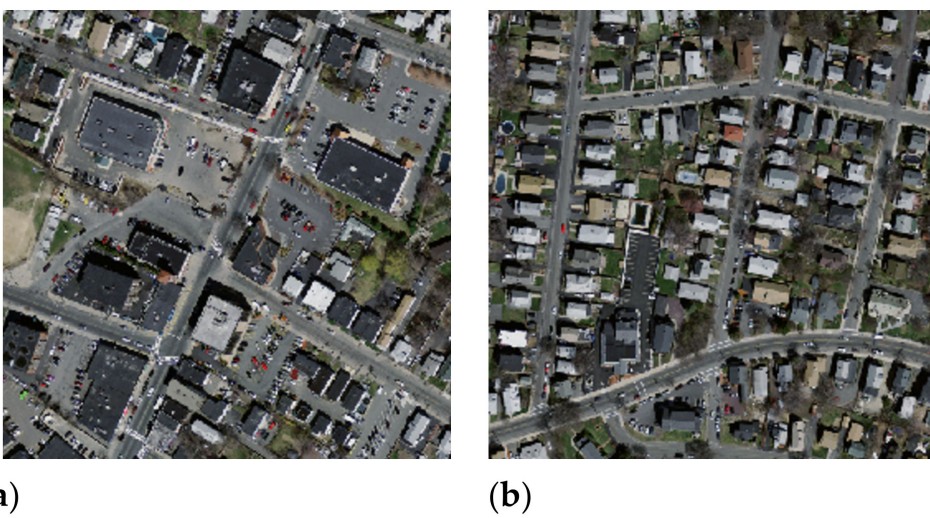

(**a**)      (**b**)

**Figure 6.** These are the two characteristics of the Massachusetts Buildings Dataset: (**a**) The diversity of this building style; (**b**) the density of this building distribution.

The WHU Satellite Dataset I (global cities) is not collected by a single satellite system. It is collected by researchers from cities around the world and various remote sensing resources, including QuickBird, worldview series, IKONOS, and ZY-3. In addition to the

differences of satellite sensors, atmospheric conditions, panchromatic, and multispectral fusion algorithms, atmospheric correction, radiometric correction, and seasons are inconsistent. This dataset contained 204 512×512 images with resolutions ranging from 0.3 to 2.5 m. We divided the dataset into training and test sets at a ratio of 153:51. Because of the hardware limitation, we cut it in the form of sliding window. After clipping, the training set contained 3825 256×256 images, and the test set contained 1275 256×256 images. The biggest advantage of WHU Satellite Dataset I (global cities) is that it has a wide range of sources, with the textures and shapes of various styles of surface buildings. It is suitable but challenging for testing the robustness of building extraction algorithms. We used it to verify the generalization ability of our method.

*4.2. Evaluation Metrics*

In the field of semantic segmentation, there are many evaluation metrics used to measure pixel classification. Frequency-weighted intersection over union (*FWIoU*), pixel accuracy (*PA*), overall accuracy (*OA*), intersection over union (*IoU*), weight parameter, and *F-score*. However, the focus of this article is building division. Essentially building segmentation is a pixel point binary classification issue rather than a multiclassification problem. Therefore, we chose several evaluation metrics to verify the strengths and weaknesses of the network model.

The validation metrics mentioned above are actually composed of four basic metrics. These four basic indicators are true positive (*TP*), true negative (*TN*), false-negative (*FN*), and false-positive (*FP*) cases. In the building segmentation task, the labeled and predicted values of the pixel points are the numbers of samples of the building category from which the *TP* value is derived. *TN* is the sample whose pixel point label value is the background category and is likewise predicted to be the background category. The value of *FN* is the number of building pixel points that are predicted to be the background class. The value of FP is the number of background pixel points that are predicted to be buildings. These four fundamental criteria are used to evaluate the success of semantic segmentation approaches.

Next, we discuss the evaluation metrics that were utilized to determine effectiveness. The first metric we chose was the *F1-score*. However, before talking about the *F1-score*, it is important to introduce two other metrics: precision and recall.

Precision is also known as the accuracy rate. In the building segmentation task, this evaluation metric can represent the probability that the label value is the pixel point of a building being successfully predicted as a building, overall predictions that result in a building pixel points. The calculation of precision is shown in Equation (12).

$$Precision = \frac{TP}{TP + FP} \tag{12}$$

Recall is also referred to as the champerty rate. This evaluation metric can represent the probability that a label value is a pixel point of a building being successfully predicted as a building, overall label values of building pixel points. The calculation of recall is shown in Equation (13).

$$Recall = \frac{TP}{TP + FN} \tag{13}$$

These two formulas illuminate the principles of the aforementioned indicators. Ideally, each researcher should design a network architecture that achieves both metrics as well as possible. However, in general, these two metrics cannot achieve high values at the same time. Sometimes, a high precision causes a lower recall and a higher recall causes a lower precision. Therefore, the *F-score* was used to consider the summation value of precision and recall together. The calculation of the *F-score* is shown in Equation (14).

$$F - Score = \left(1 + \beta^2\right) \frac{Precison \times Recall}{(\beta^2 \times Precision) + Recall} \tag{14}$$

In this equation, the $\beta$ is a hyperparameter that regulates the relationship between precision and recall, and the meaning of the formula depends on taking different values. The evaluation index chosen in this paper was the *F1-score*, which means that $\beta$ took the value of 1. This represents the reconciled average evaluation metric of precision and recall. The calculation of the *F1-score* is shown in Equation (15).

$$F1 - Score = \frac{2 \times Precison \times Recall}{Precision + Recall} \qquad (15)$$

The second evaluation metric chosen in this paper was *IoU*, which is calculated as the ratio of the intersection of predicted and true values. The calculation of *IoU* is shown in Equation (16).

$$IoU = \frac{TP}{FN + TP + FP} \qquad (16)$$

The third evaluation metric we chose is *OA*. *OA* is the ratio of the number of correctly predicted samples to the overall sample size. The calculation of *OA* is shown in Equation (17)

$$OA = \frac{TP + TN}{FN + TP + FP + TN} \qquad (17)$$

GFLOPs means giga floating-point operations per second, that is, one billion floating-point operations per second. GFLOPs is often used as a GPU performance parameter, and it was used here to evaluate the complexity of the model. The other metrics were used to evaluate the size of the model, and they were obtained in the Keras framework through fixed code.

### 4.3. Implementation Details

The experimental environment was Windows10. The CPU model was Intel(R) Core (TM) i7-10700 CPU @ 2.90GHz, and the GPU model was 2080Ti. Because of the hardware limitations in the lab, the network model could not directly load 1500×1500 resolution images in this experiment. The data processing strategy used in this experiment was sliding window cropping, which performed both horizontal and vertical cropping. This is also a method of data expansion. In our experiments, we mainly used the Keras framework, and the parameters of the initialized network model were all consistent. We employed Adam as the optimizer during the training phase, with a 1e-4 starting learning rate. The training batch size was set to 4. We also set some training strategies through Keras: if the loss remains unchanged after three epochs of training, the network model automatically modifies the learning rate to half of the original one; if the loss does not change after ten epochs of training and learning rate modification, the network model automatically stops the training process and saves the final network weights.

### 4.4. Comparisons and Analysis

In this section, we compare and contrast our model with classical and advanced network models. First of all, the models are as follows:

- FCN [25] was the first deep learning semantic segmentation network. This network model can be used to extract network features through a CNN backbone network, and then the features are shrunk to the same proportions as the original image through several deconvolution layers. This end-to-end approach has seen very significant advances in the data-driven era.
- UNet [29] was recently proposed, mainly for biomedical image segmentation. Nowadays, many human tissue segmentation networks in the field of medical image segmentation use UNet as their main framework. Building extraction and medical image segmentation usually have the same goal of pixel point binary classification. Moreover, UNet was the first model to utilize a network of encoding and decoding modules for semantic segmentation. UNet views the process of acquiring a high semantic feature

map from an image as an encoder and the process of obtaining a high semantic feature map from a pixel-level mask map as a decoder. Segmentation performance can be effectively ensured by complementing the convolution information needed for the decoder's deconvolution process with the encoder's detailed information.

- SegNet [27], also based on FCN, is a coding and decoding network like UNet; SegNet uses VGG-16 as its backbone network. However, the training convergence of SegNet is faster than that of the UNet network due to this pooling index.
- GMEDN [12] is a network specifically designed to perform building extraction. This network adds modules such as attention, knowledge distillation, and feature fusion modules to make the network better at extracting spatial multi-scale information.
- BRRNet [51] uses a residual refinement network for post-processing; this network can effectively refine the edges of buildings and improve accuracy.

The Massachusetts Building Dataset was fitted to each selected model to obtain the results presented in Table 1. The effect of SSDBN was very distinct. Our proposed network received the highest marks for precision, accuracy, *IoU*, *F1-score*, weight parameters, model size, and GFLOPs among the six networks. Compared to the other five networks, the minimum improvement of our SSDBN on *IoU* was 1.1%, the minimum improvement on *F1-score* was 1%, the minimum number of parameters was reduced by 46.58 MB, the minimum model size was reduced by 12.24 MB, and the GFLOPs was reduced by 17.6. We also added training time, the time spent in each epoch during training, as an evaluation metric. There were 7488 images per training, and the batch size was set to 4. Because the test time was proportional to the training time, we did not list the test time. From Table 1, we can see that SSDBN took the shortest time for training. A comparison of several other previous metrics showed that our SSDBN had the best performance.

**Table 1.** Comparison of different models with the Massachusetts Building Dataset.

| Model Name | Massachusetts | | | | | | |
|:---:|:---:|:---:|:---:|:---:|:---:|:---:|:---:|
| | IoU | OA | F1-Score | Parameters | Model Size | GFLOPs | Time |
| FCN | 0.6650 | 92.66% | 0.7994 | 512.27 M | 134.35 M | 55.42 | 383 s |
| UNet | 0.7311 | 93.63% | 0.8372 | 95.03 MB | 24.89 M | 112.63 | 200 s |
| SegNet | 0.6891 | 93.56% | 0.8222 | 112.44 MB | 29.44 M | 40.14 | 242 s |
| GMEDN | 0.7473 | 94.72% | 0.8668 | 725.34 MB | 190.11 M | 68.41 | 483 s |
| BRRNet | 0.7446 | 94.61% | 0.8536 | 66.38 MB | 17.35 M | 117.39 | 373 s |
| SSDBN | 0.7583 | 95.35% | 0.8769 | 19.8 MB | 5.11 M | 22.54 | 172 s |

The FCN, which is the original proposed deep learning network is suitable for all fields, was the least effective of these network models because precision generally increase reverse while universality decreases. Multi-scale remote sensing data and the imbalance of positive and negative samples were not well-considered by the FCN, so it had the worst effect on this dataset. SegNet had a better up-sampling strategy, so the overall performance was better than that of FCN. UNet, on the other hand, greatly improved performance by fusing high-dimensional semantic and low-dimensional detailed information. The BRRNet had a simple structure but improved the performance by post-processing the residual units, thus demonstrating a better performance than the first three networks. The GMEDN improved performance by adding many plug-and-play modules, such as those of attention mechanism and knowledge distillation. Although the overall network performance was improved, the number of network model parameters was super high; the authors of the GMEDN sacrificed real-time computation to improve performance.

We randomly selected some cropped images of only 256 × 256 size from the Massachusetts Building Dataset for prediction. Figure 7 illustrates the results regarding the abovementioned quantitative metrics, for a selection of prediction images from the visualization results of the dataset.

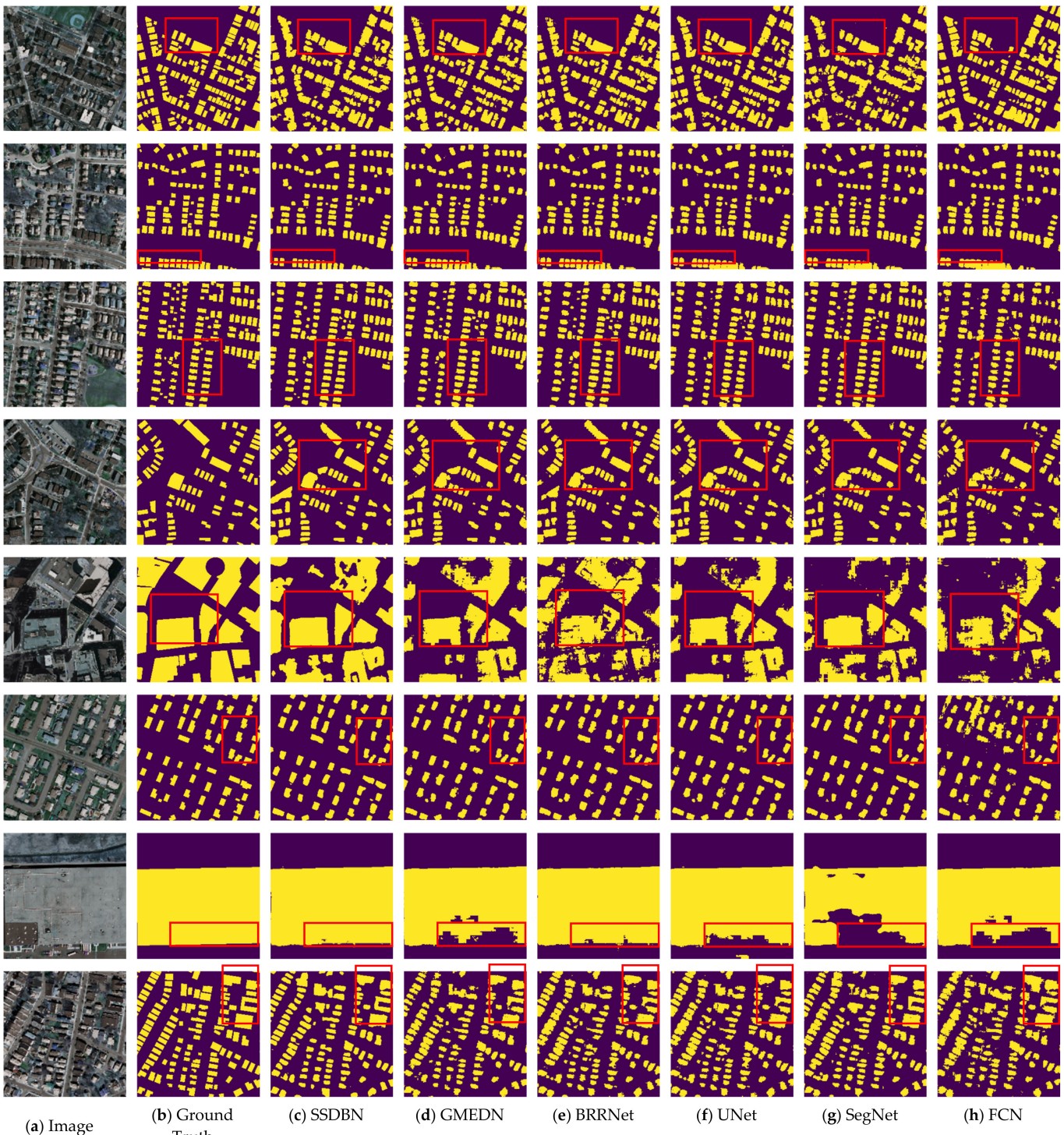

|  (**a**) Image | (**b**) Ground Truth | (**c**) SSDBN | (**d**) GMEDN | (**e**) BRRNet | (**f**) UNet | (**g**) SegNet | (**h**) FCN |

**Figure 7.** Prediction of different models on the Massachusetts Building Dataset. (**a**) Input image; (**b**) ground true; (**c**) the result of our method of SSDBN; (**d**) the result of GMEDN; (**e**) the result of BRRNet; (**f**) the result of UNet; (**g**) the result of SegNet; (**h**) the result of FCN. The red boxes indicate the more obvious extraction results.

In Figure 7, the obvious improvements are marked with red boxes. One can see that SSDBN, BRRNet, and GMEDN worked well when the density of small buildings was high. However, UNet, SegNet, and FCN did not work well. When predicting large buildings, BRRNet, SSDBN, and UNet were better at preserving geometric details and contours. Overall, SSDBN was better at capturing multi-scale information than the other networks.

*4.5. Ablation Analysis*

4.5.1. Ablation between Modules

Our proposed SSDBN model consists of different blocks used to capture information at multiple scales. To demonstrate the effectiveness of these modules for the final results, we designed an ablation experiment in which we designed five groups of networks. Table 2 shows the result of ablation experiments.

**Table 2.** Ablation study of our SSDBN.

| Network | IoU | F1-Score |
| --- | --- | --- |
| Net1 | 0.7583 | 0.8769 |
| Net2 | 0.7406 | 0.8676 |
| Net3 | 0.7313 | 0.8589 |
| Net4 | 0.7002 | 0.8583 |
| Net5 | 0.7401 | 0.8712 |

- Net1: SSDBN.
- Net2: Compared to Net1, Net2 does not have a central block.
- Net3: Compared to Net1, the Net3's backbone network uses the original Xception.
- Net4: Compared to Net1, Net4 uses a common decoder.
- Net5: Compared to Net1, Net5 does not have an attention module.

Through the experimental comparisons, we obtained the results in Table 2. In order to exclude some other factors, *IoU* and *F1-score* were chosen for the evaluation index when conducting the ablation experiment. Each network was compared with Net1(SSDBN), and we found that each module contributed to the final evaluation metric score.

Specifically, when the central block was removed, IoU was 1.77% lower than SSDBN, and when the backbone network used the original, *IoU* was 2.7% lower than SSDBN; the *F1-score* was also inferior to SSDBN. When we did not use the attention module, the *IoU* and *F1-score* ere not improved. When using the ordinary decoder, the *IoU* and *F1-score* were reduced by 5.81% and 1.86%, respectively, compared to our SSDBN. Therefore, it could be seen that the dual-branch decoder we designed played a key role in the performance improvement of this model. As soon as any of these modules were missing, the overall network evaluation metric dropped a little. These modules led to the high performance of the SSDBN.

4.5.2. Ablation inside the Res2Net

The authors of this paper modified Res2Net to enhance the extraction of deep multi-scale information. Therefore, in this section, we compare many Res2Net variants to demonstrate the superiority of our modified structure. Table 3 shows the result of ablation experiments inside the Res2Net.

**Table 3.** Ablation study of our Res2Netplus.

| Network | IoU | F1-Score |
| --- | --- | --- |
| Net1 | 0.7583 | 0.8769 |
| Net6 | 0.7275 | 0.8606 |
| Net7 | 0.7411 | 0.8682 |
| Net8 | 0.7391 | 0.8710 |
| Net9 | 0.7379 | 0.8691 |

- Net1: SSDBN.
- Net6: Compared to Net1, Net6 is divided into five parts by channel when using Res2Netplus.
- Net7: Compared to Net1, Net7 is divided into three parts by channel when using Res2Netplus.

- fNet8: Compared to Net1, Net8 makes use of ordinary convolution when using Res2Netplus.
- Net9: Compared to Net1, Net9 uses the baseline Res2Net.

In order to prove the effectiveness of our improved Re2Netplus, we performed ablation experiments. A comparison of the advantages and disadvantages of the Res2Net variants can be seen in Table 3. When we divided the feature map into five groups by channel in the ablation experiment, we obtained the worst values of *IoU* and *F1-score*. When we divided it into three groups, the *IoU* and *F1-score* increased to 0.7411 and 0.8682, respectively, but still did not perform well in the four groups. In addition, we used ordinary convolution in our designed Re2Netplus and compared it with Net1 (Re2Netplus), and we found that the effect was not as good as our dilated convolution. Finally, we used the original Res2Net as the baseline model, and we found that the IoU and *F1-score* of our designed Res2Netplus were increased by 2.04% and 0.78%, respectively. This proves that our improvements were positive.

After improving this Res2Net, we found that our proposed Res2Netplus worked well. The enhancement of Res2Netplus to fit the nature of remote sensing images due to modifications of Res2Net is evident.

### 4.6. Generalization Ability

WHU Satellite Dataset I (global cities) was used to verify the generalization ability of our method. We compared SSDBN with the most advanced method, as shown in Table 4 and Figure 8.

**Table 4.** Comparison of different models with the WHU Satellite Dataset I (global cities).

| Model Name | WHU Satellite Dataset I (Global Cities) | |
| --- | --- | --- |
| | **IoU** | **F1-Score** |
| FCN | 0.5105 | 0.6841 |
| U-Net | 0.7192 | 0.8269 |
| SegNet | 0.6705 | 0.8126 |
| GMEDN | 0.7104 | 0.8238 |
| BRRNet | 0.7066 | 0.8181 |
| SSDBN | 0.7200 | 0.8273 |

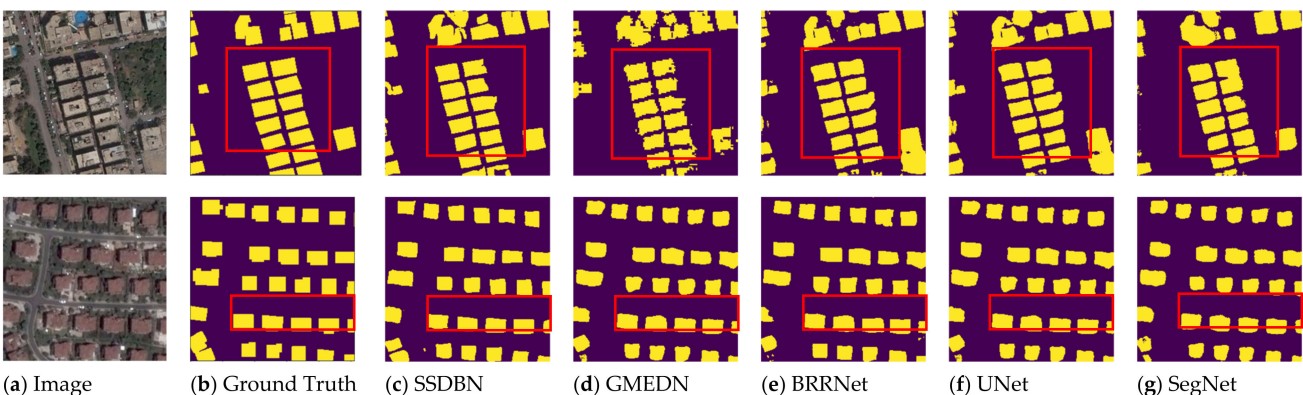

(**a**) Image    (**b**) Ground Truth    (**c**) SSDBN    (**d**) GMEDN    (**e**) BRRNet    (**f**) UNet    (**g**) SegNet

**Figure 8.** Prediction of different models with the WHU Satellite Dataset I (global cities). (**a**) Input image; (**b**) ground true; (**c**) the result of our method of SSDBN; (**d**) the result of GMEDN; (**e**) the result of BRRNet; (**f**) the result of UNet; (**g**) the result of SegNet. The red boxes indicate the more obvious extraction results.

From the comparison between SSDBN and the five selected state of the art methods in Table 4, it can be seen that both *IoU* and *F1-score* indicators were optimal; SSDBN presented 20.95% and 14.32% improvement of *IoU* and *F1-score*, respectively, compared to the FCN. In terms of quantitative results, SSDBN was only 0.08% better than UNet (0.04% higher *IoU* and *F1-score* values), but our advantages were obvious compared to other improved methods such as SegNet, GMEDN, and BRRNet. On the whole, our SSDBN model showed positive significance regarding generalization ability.

From the red boxes in Figure 8, we can clearly see that our model had the best building extraction ability. On this small dataset, none of the results were particularly good, though, but with limited training resources, our model was able to reach a level ahead of others. Compared to other methods, there were less problems of error extraction, which further proves that our model is applicable to other datasets.

## 5. Conclusions

Based on high-resolution remote sensing images, a new, useful, single-side double-branch coding and decoding network model (SSDBN) was designed for this paper. The authors of this paper mainly improved the multi-scale problem of ground objects in remote sensing high-resolution images.

1.  In order to extract global and local context details from remote sensing images, building information can be fully extracted. In the decoder stage, the Xception network was used as the backbone network. The improved Re2Netplus unit could obtain deep multi-scale semantic information in the image. The CBAM attention mechanism was used to obtain the weight of characteristic pixels. These modules could make the network more comprehensive for local and non-local information extraction than previously used methods.
2.  In order to expand the receptive field of feature extraction, we used hole convolution in the feature module and different dilated rates to extract a larger receptive field in order to realize feature information fusion.
3.  In order to capture high-level semantic information and multi-scale information at the same time, as well as fully learn the low-level and high-level visual features, we designed a deconvolution integral branch and a feature enhancement branch in the decoder stage. The deconvolution branch was mainly used to capture the low-level and high-level basic feature information of buildings and supplement potential semantic information. In the feature enhancement branch, a jump connection was used to further enhance semantic information and multi-scale information, as well as supplement the final output mapping.

SSDBN was found to realize building extraction for the Massachusetts Building Dataset and WHU Satellite Dataset I (global cities). Its performance was proven in various experiments. Compared with the most advanced methods, for Massachusetts, the minimum improvements of our SSDBN were 1.1% for *IoU*, 0.63% for *OA*, and 1% for *F1-score*. At the same time, the parameters and model size of the SSDBN are light weight, which is appropriate in terms of time and space complexity. For WHU Satellite Dataset I (global cities), the minimum improvements of our SSDBN were 0.08% for *IoU*, 0.04% for *F1-score*, 17.6% (reduction) for GFLOPs. Although our SSDBN achieved good results in building extraction tasks, in some cases, the extraction ability for building edges was poor because of shadows caused by illumination. When some ground textures are similar to buildings, false recognition occurs. In order to solve this problem, we will consider object-based context structure to improve this problem in the future.

**Author Contributions:** Q.L. designed the project, oversaw the process, and improved the manuscript; Y.L. and H.L. proposed the methodology, completed the programming, and wrote the manuscript; Y.Z. and X.L. analyzed the data. All authors have read and agreed to the published version of the manuscript.

**Funding:** This work has received funding from the Key Laboratory Foundation of National Defence Technology under Grant 61424010208, National Natural Science Foundation of China (No. 62002276, 41911530242 and 41975142), 5150 Spring Specialists (05492018012 and 05762018039), Major Program of the National Social Science Fund of China (Grant No. 17ZDA092), 333 High-Level Talent Cultivation Project of Jiangsu Province (BRA2018332), Royal Society of Edinburgh, UK and China Natural Science Foundation Council (RSE Reference: 62967\_Liu\_2018\_2) under their Joint International Projects funding scheme and basic Research Programs (Natural Science Foundation) of Jiangsu Province (BK20191398 and BK20180794).

**Data Availability Statement:** Publicly available datasets were analyzed in this study. Massachusetts Buildings Dataset can be found here: [https://www.cs.toronto.edu/~vmnih/data/](https://www.cs.toronto.edu/~vmnih/data/) (accessed on 5 December 2021)]. The WHU dataset presented in this study are available in ref [50].

**Conflicts of Interest:** The authors declare no conflict of interest. The funders had no role in the design of the study; in the collection, analyses, or interpretation of data; in the writing of the manuscript, or in the decision to publish the results.

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
