# Peer review of "SSDBN: A Single-Side Dual-Branch Network with Encoder–Decoder for Building Extraction"

_remotesensing, doi:10.3390/rs14030768_

Round 1

Reviewer 1 Report

Authors propose a CNN based architecture for building image extraction. While doing so they have considered varied structure and size of the buildings as well as generating the enhanced segmented features using Res2Net, xception, and CBAM based architecture. A detailed ablation study has been provided to prove the validity of the modules used in the proposed dual decoder architecture. 

Comments:

--Figure 1 should be updated with the detailed arrows direction from encoder-to-decoder.

--Author use hole/hollow convolution in order to form Res2Netplus form ResNet. However, a brief overview of what hole/hollow convolution is about is missing. 

--In line 255 "cavity convolution" term has been coined for a single time in the paper. What does that refer?

-- Regarding english language, I feel a detailed and extensive rewriting of many sentences and text as a whole is required. In line 301, it has been glorified the earlier contribution with the phrases like "great sensation in the academic world", "great creation", etc. Rather than using such phrases, it is always better to explain the benefits of those methods and how do they be advantageous in the context of this work. Further, Some sentences are quite long and redundant. For example: line 393-396.

--Explanation for the existing methods like 'CBAM'  are quite long. Since these are already established methods and are not the contribution of the paper, it is enough to give a brief overview rather than spending several pages.

--Authors claim that the use of Res2NetPlus generates enhanced features compared to Res2Net, the performance of the proposed network when using one to another should be analyzed in detail. Quantitative results to compare the performances are desired.

Author Response

Point 1: Figure 1 should be updated with the detailed arrows direction from encoder-to-decoder.

Response 1: According to your suggestion, in Figure 1, we have remade this diagram, pointing the arrow more clearly and explaining the symbolic operators.

Point 2: Author use hole/hollow convolution in order to form Res2Netplus form ResNet. However, a brief overview of what hole/hollow convolution is about is missing.

Response 2: According to your suggestion, we added the introduction of dilated convolution in line233-line240 in Section 3.2.

Point 3: In line 255 "cavity convolution" term has been coined for a single time in the paper. What does that refer?

Response 3: Thank your question, the “cavity vonvolution” refers to dilated convolution. According to your suggestion, here, we have collectively referred to this name as dilated convolution.

Point 4: Regarding english language, I feel a detailed and extensive rewriting of many sentences and text as a whole is required. In line 301, it has been glorified the earlier contribution with the phrases like "great sensation in the academic world", "great creation", etc. Rather than using such phrases, it is always better to explain the benefits of those methods and how do they be advantageous in the context of this work. Further, Some sentences are quite long and redundant. For example: line 393-396.

Response 4: Thank your question, according to your suggestions, we have rewritten the sentence to clearly describe the advantages of this method. Besides, we have deleted the sentences.

Point 5: Explanation for the existing methods like 'CBAM'  are quite long. Since these are already established methods and are not the contribution of the paper, it is enough to give a brief overview rather than spending several pages.

Response 5: Thank you for your suggestion. In According to your suggestion, we deleted the relevant introduction of CBAM and summarized CBAM as a whole. Only one module diagram describing the complete CBAM is retained, and the formula is retained in order to help readers better understand its working principle.

Point 6: Authors claim that the use of Res2NetPlus generates enhanced features compared to Res2Net, the performance of the proposed network when using one to another should be analyzed in detail. Quantitative results to compare the performances are desired.

Response 6: Thank you for your suggestion. In fact, we have conducted ablation experiments on our designed Res2Netplus in section 4.5.2 of this paper. We have designed five groups of experiments, which are to divide the characteristic diagram into 3, 4 and 5 groups according to the channel, and use ordinary convolution and hole convolution to reduce the difference between the results. We analyze the advantages of our designed Re2Netplus from the quantitative results and intuition In order to more clearly describe the advantages of this method, we re analyze and discuss in detail based on the quantitative results in Table 3.

Reviewer 2 Report

The paper presents a new two-branch network aiming to solve the multi-scale problem of building objects from high-resolution satellite images. The proposed network contains three main parts: decoder module, feature fusion module, and two-branch decoder module. Decoder and Xception are used as backbone network to acquire deeper multi-scale semantic information in satellite image through an improved Res2Net unit. The experiments showed that the proposed network can achieve better results than the state-of-the-art approaches. On the other hand, it also showed high efficiency for the detection process.

In overall, the innovation of this paper is significant. However, the presentation of this paper is really bad. In many places, explanations to algorithms are missing or not sufficient. This makes the paper hard to be understood. Furthermore, the language of this paper needs to be polished by someone whose English is better.

The detailed comments are listed as follows:

  1. Why is the proposed network called single-side?
  2. What is the problem to be solved in this work? Please point this out explicitly in the introduction and explain clearly how the proposed method can solve the problem.
  3. The authors referred to a data imbalance problem in deep learning. Did you solve this problem in the presented work? Where and how?
  4. Page 2, line 72: many methods proposed later are based on the FCN itself. What is the problem with this fact?
  5. The last sentence in the first point of contribution should be removed, since this is an introduction.
  6. A brief introduction of the structure of the paper is missing in the introduction section.
  7. What is the conclusion of the related work section?
  8. Page 6, line 224 to 225: in the experiment, we found that dividing into four groups is the best for the whole experiment effect. What does this mean? How much data was used? How many experiments were conducted? Does this mean that the proposed method is data-dependent?
  9. In Figure 9, please mark where are the good results and bad results of your method in comparison to others. Where and what are difference in the results?
  10. Page 16, the first paragraph in Section 4.5.1 is duplicated with the one after the table 2 in the same subsection.
  11. More detailed explanation for Table 2 and 3 is needed.
  12. The conclusion needs to be rewritten. It does not appear like a conclusion.
  13. What are limitations of the proposed method?

Author Response

Point 1: Why is the proposed network called single-side?

Response 1: Thank you for your question. The network we proposed is called single-side dual branch network because the overall network idea is the idea of encoder and decoder, which can be regarded as a U-shape, with encoder on one side and decoder on the other. When we designed the network, we also designed a bilateral dual branch network (both encoder and decoder have dual branches), and the result is not ideal. Therefore, at present, our single-side dual branch only makes use of the dual branch on the decoder side.

Point 2: What is the problem to be solved in this work? Please point this out explicitly in the introduction and explain clearly how the proposed method can solve the problem.

Response 2: Thank you for your suggestions. We have stated the problems to be solved and the methods to be used in this article in line85-line90 of the introduction, but we have restated this part for clearer expression.

“This decoder network transfers the features extracted by the center module to the two branches of the decoder respectively and further improves the accuracy and details of building extraction by correcting the residuals between the main branch of the decoder network and the feature map.” changed to “ The decoder network transmits the features extracted by the central module to the deconvolution and feature enhancement branches of the decoder respectively, captures multi-scale information and high-level semantic information respectively, and further improves the accuracy and detail of building extraction by correcting the residual between the main branch of the decoder network and feature mapping.”

Point 3: The authors referred to a data imbalance problem in deep learning. Did you solve this problem in the presented work? Where and how?

Response 3:Thank you for your suggestion, In this paper, the imbalance of building segmentation data set is mainly divided into two parts:

  1. Data processing. In the data processing part, because of the limitations of the graphics card, we must slide and cut the pictures before we can put them into the network for training. After cutting a 256 × 256 image, negative samples will appear in the data set, that is, the whole 256 × 256 images is background pixels. After manual screening, we can remove some negative samples to balance the positive and negative samples. The data set production and processing are mainly described in section 4.1.
  2. Loss function. The loss function dice loss is used to alleviate the data imbalance in the semantic segmentation task, so as to improve F1-Score. This is mainly the conclusion put forward by predecessors. We did not put forward innovation in the loss function, so we did not spend a lot of space to describe it in this paper.

Point 4: Page 2, line 72: many methods proposed later are based on the FCN itself. What is the problem with this fact?

Response 4: Thank you for your suggestion, in line 71- line 72, we restate this sentence and cite relevant references based on the method proposed by FCN for proof. Through this sentence, we want to draw a conclusion that although FCN is widely used, it also has shortcomings.

Point 5: The last sentence in the first point of contribution should be removed, since this is an introduction.

Response 5: Accoding to your suggestion, we have removed the last sentence in the first of contribution.

Point 6: A brief introduction of the structure of the paper is missing in the introduction section.

Response 6: Thank you for your suggestion. In line 111 -line 115, we have supplemented the arrangement of the full-text organizational structure at the end of this section.

Point 7: What is the conclusion of the related work section?

Response 7: Thank you for your suggestions. In line 171 -line 179, we have supplemented the conclusions at the end of the second part of the related work。

Point 8: Page 6, line 224 to 225: in the experiment, we found that dividing into four groups is the best for the whole experiment effect. What does this mean? How much data was used? How many experiments were conducted? Does this mean that the proposed method is data-dependent?

Response 8: This sentence of “in the experiment, we found that dividing into four groups is the best for the whole experiment effect.” is proposed for res2net. Because there is a step in res2net to divide the incoming feature map into four parts according to the channel. Later, in the ablation experiment, we also compared with res2net (divided into five parts by channel) and Res2Net (divided into three parts by channel). On the mass data set, we found that the effect of dividing into four parts by channel is the best.

Besides, we use two datasets introduced in Section 4.1 to carry out experiments to prove that the proposed method does not depend on a specific data.

Point 9: In Figure 9, please mark where are the good results and bad results of your method in comparison to others. Where and what are difference in the results?

Response 9: Thank you for your suggestion, we have maked the where are the good result and bad results in Figure 9.

Point 10: Page 16, the first paragraph in Section 4.5.1 is duplicated with the one after the table 2 in the same subsection.

Response 10: Thank you for your suggestion, we have deleted the duplicated paragraph.

Point 11: More detailed explanation for Table 2 and 3 is needed.

Response 11: According to your suggestion, we supplement the discussion in Table 2 and table 3. Table 2 compares different evaluation index values of the model based on our proposed method, and obtains the advantages of our method in combination with the current advanced methods. In Table 3, through ablation experiments, the contributions of different modules of our proposed method are compared from the evaluation indexes, and it is concluded that the design of two branches in the decoder stage of our proposed method plays a key role.

Point 12: The conclusion needs to be rewritten. It does not appear like a conclusion.

Response 12: According to your suggestion, we have rewritten the conclusion.

Point 13: What are limitations of the proposed method?

Response 13: Thank you for your question, although our method improves the edge of building extraction, it can still be further studied to make it smoother. And the extraction accuracy in multiple scenes needs and the generalization of the model to be further improved.Although our method shows good extraction effect after a large number of tests, in some cases, the extraction ability of building edge is poor because of shadows caused by illumination. When some ground textures are similar to buildings, false recognition will occur. Therefore, we consider object-based context structure to improve this problem in the future.

Reviewer 3 Report

The paper proposes and experiment SSDBN as a new two-branch network as a solution for building classification in remote sensing high-resolution images. The experiments are based on the Massachusetts Building Dataset. The proposed neural network is compared with a well known neural networks from the literature.

The first mention of the abbreviation CNN should be associated with its meaning Convolutional Neural Network. The same observation on the abbreviation VGG network, CBAM. They must be explained at the first mention.

In the line 229 "i represents the number of channels" or "channel number", "channel index"?
In the line 231 I think that "subset xi and Ki-1(xi) is added together" has to be "?i-1(xi-1)"
In line 274 I do not understand the expression ??(??+??−1) that seems to be different that figure 2. The correct one seems to be Ki(xi)+Ki-1. 
Explain expression (3) C=concat(x1,K2,K3,K4). What means concat?  

The text in the lines 278-282 is quite confuse. Reformulate! What do you understand by "clearly conclude", "very clear"? Is it an intuitive conclusion or is it based on a metric?  

Reformulate "Imagine that the input image input size" in line 284.
In line 288 the formula (4) describes formally a concept rather than a mathematical expression. Detail the explanation. What is the meaning of "learning weight"?

I do not understand the meaning of the following expression "the receptive field is far from insufficient". Is it sufficient? Please reformulate.

Explain the symbolic operators in Fig.1, 4 and 5.

Reformulate for more understanding "Since our proposed SSDBN model was added quite a lot of modules to capture information at multiple scales."

The text is repeated in section 4.5.1 "Since our proposed SSDBN model was added quite a lot of modules to capture multi-scale information. To demonstrate the effectiveness of these modules for the final results, so we designed this ablation experiment."

Some editing mistakes:
- "suing"
- the text "Although these proposed methods have achieved good results in the extraction of buildings from remote sensing images, they also exposed some problems." is doubled in the same paragraph.
- the following expression "High-resolution remote sensing images have higher resolution than general images and" is obvious and needs to be reformulated.
- capitalise the words "Massachusetts building dataset" to "Massachusetts Building Dataset"
- explain the acronim CRF. What is FC layer?
- reformulate "an intermediate FC layer as an intermediary"
- "Feture Fusion module" in Fig.1
- "original Res2net module"
- is missing a dot in "convolution The features"
- ???−???????????
- in the line 475 "F1score, which means that x takes", should be f1-score, and beta instead of x.
- . . .

Author Response

Point 1: The first mention of the abbreviation CNN should be associated with its meaning Convolutional Neural Network. The same observation on the abbreviation VGG network, CBAM. They must be explained at the first mention.

Response 1: Thank you for your suggestion. We have carefully checked the full text and supplemented all the abbreviations of academic terms proposed for the first time. Specifically,

Convolutional Neural Network - CNN

Fully Convolutional Network - FCN

Binary Cross Entropy - BCE

Visual Geometry Group Network 16 - VGG 16

Conditional Random Field - CRF

Fully Connected - FC

Convolutional Block Attention Module - CBAM

Point 2: In the line 229 "i represents the number of channels" or "channel number", "channel index"?

Response 2: Thank you for your suggestion. We have re expressed the meaning of I, which really represents the channel index.

Point 3: In the line 231 I think that "subset xi and Ki-1(xi) is added together" has to be "?i-1(xi-1)"

Response 3: Thank you for your suggestion. The description of this sentence is ambiguous. Therefore, we use yi-1 directly instead of it.

Point 4: In line 274 I do not understand the expression ??(??+??−1) that seems to be different that figure 2. The correct one seems to be Ki(xi)+Ki-1.

Response 4: Thank you for your suggestion. Your opinion is correct and we have corrected it.

Point 5: Explain expression (3) C=concat(x1,K2,K3,K4). What means concat?

Response 5: According to your suggestion, we explained the meaning of concat, which is expressed as: the addition of the features obtained from the four groups of feature maps x_1, K_2, K_3, K_4 after dilated convolution.

Point 6: The text in the lines 278-282 is quite confuse. Reformulate! What do you understand by "clearly conclude", "very clear"? Is it an intuitive conclusion or is it based on a metric?

Response 6: According to your suggestion, we illustrate that our improvement is effective from the quantitative indicators and intuitive aspects of the later test. Details are described in line 308-line 313.

Point 7: Reformulate "Imagine that the input image input size" in line 284.

Response 7: Thank you for your suggestion. In line 316, we have revised this sentence, specifically : “Suppose the size of the input image” modified to “Imagine that the input image input size”.

Point 8: In line 288 the formula (4) describes formally a concept rather than a mathematical expression. Detail the explanation. What is the meaning of "learning weight"?

Response 8: Thank you for your suggestion. In line 323 -325, we have explained "learning weight", which is specifically expressed as “The weight can be understood as the importance of the control signal. In this paper, it represents the important pixels expressing building information in the deep network learning remote sensing image. The weight changes until the best value is learned.”

Point 9: I do not understand the meaning of the following expression "the receptive field is far from insufficient". Is it sufficient? Please reformulate.

Response 9: Thank you for your suggestion. We restated this sentence in line271-line274. Specifically:“the receptive field is far from insufficient”modified to “ the receptive field of ordinary convolution kernel is not open enough to cap-ture deeper information”.

Point 10: Explain the symbolic operators in Fig.1, 4 and 5.

Response 10: Thank you for your suggestion. We have explained the symobolic operators in all Figures.

Point 11: Reformulate for more understanding "Since our proposed SSDBN model was added quite a lot of modules to capture information at multiple scales."

Response 11: Thank you for your suggestion.We have reformulated the sentence. Sepcifically, "Since our proposed SSDBN model was added quite a lot of modules to capture information at multiple scales.",changed to “Our proposed SSDBN model consists of different blocks to capture information at multiple scales.

Point 12: The text is repeated in section 4.5.1 "Since our proposed SSDBN model was added quite a lot of modules to capture multi-scale information. To demonstrate the effectiveness of these modules for the final results, so we designed this ablation experiment."

Response 12: Thank you for your suggestion. We have deleted the repeated text.

Point 13:  Some editing mistakes:

- "suing"

- the text "Although these proposed methods have achieved good results in the extraction of buildings from remote sensing images, they also exposed some problems." is doubled in the same paragraph.

- the following expression "High-resolution remote sensing images have higher resolution than general images and" is obvious and needs to be reformulated.

- capitalise the words "Massachusetts building dataset" to "Massachusetts Building Dataset"

- explain the acronim CRF. What is FC layer?

- reformulate "an intermediate FC layer as an intermediary"

- "Feture Fusion module" in Fig.1

- "original Res2net module"

- is missing a dot in "convolution The features"

- ???−???????????

- in the line 475 "F1score, which means that x takes", should be f1-score, and beta instead of x.

- . . .

Response 13: Thank you for your suggestion. I have corrected the editing mistakes of words, and checked the spelling of words in the full text and corrected them. The details are as follows:

  • In line 21, In Abstract, ‘suing’ has modified to ‘using’
  • the text "Although these proposed methods have achieved good results in the extraction of buildings from remote sensing images, they also exposed some problems." is doubled in the same paragraph. We have removed that one.
  • “High-resolution remote sensing images” changed to “High-resolution remote sensing images can clearly e xpress the spatial structure of ground objects”
  • Capitalise the words "Massachusetts building dataset" to "Massachusetts Building Dataset"
  • FC is the full name of Fully Connected.
  • “DeconvNet [28] has an intermediate FC layer as an intermediary to enhance the classification of categories.”modified to “Deconvnet [28] designed a fully connected (FC) layer in the middle as a term to strengthen category classification.”
  • We have modified the word of “Feture” of "Feture Fusion module" in Fig.1 to “Feature”.
  • Line 278, “original Res2net module" modified to “Res2Net”
  • We have added a dot in "convolution The features" in line 382.
  • In formulate 9, “de-convolution” modified to “deconvolution”
  • in the line 515 "F1score, which means that x takes", changed to F1-score, and instead of x.

Reviewer 4 Report

In this work, a Single-Side Dual-Branch Network (SSDBN) based on a 14 encoder-decoder structure is proposed. The work is good but there is a number of comments should be addressed.

  • The English writing should be revised. In the abstract, for example, the article “an” in the sentence “based on an 14 encoder-decoder structure”. Also, why you capitalize the first character of some normal words like “Dataset” in “Massachusetts Buildings Dataset”
  • The authors used encoder-decoder structure. However, the encoder-decoder structure is more sensitive to input errors. How the authors can mitigate or solve this problem.
  • Also, in the encoder-decoder structure, the compression degradation affects system performance in a significant way. How this problem can be solved.
  • The author said “a lightweight dual-branch encoder-decoder framework”. How they validate that the framework is lightweight?! I didn’t see any proof for this.
  • The authors should measure the computational efficiency in terms of execution time and time complexity notation.
  • The procedure of setup the dataset for evolution is not clear. Please explain clearly how many images for getting the results.
  • The dataset contains 151 high-resolution aerial 423 images. It is a small number for evaluation the model. Please use also another dataset with more number of images for evaluation.
  • How the authors selected the important parameters of the model and their suitable values. Please explain that.

Author Response

Point 1: The English writing should be revised. In the abstract, for example, the article “an” in the sentence “based on an 14 encoder-decoder structure”. Also, why you capitalize the first character of some normal words like “Dataset” in “Massachusetts Buildings Dataset”

Response 1: Thank you for your suggestion. We checked the full text. There is no "14" in the sentence "based on an encoder decoder structure". At the same time, according to your suggestion, we have unified the name of this dataset to “Massachusetts Buildings Dataset”, because it is a proprietary dataset, and we capitalize the initials in the full text.

Point 2: The authors used encoder-decoder structure. However, the encoder-decoder structure is more sensitive to input errors. How the authors can mitigate or solve this problem.

Response 2: Thank you for your question. The encoder decoder structure is more sensitive to input errors. Our main solution is to be more strict with the input data. In the data set of network training, testing and verification, we conduct manual screening to avoid wrong data input and obtain pure data set. Another solution is to reduce the complexity of the network, which is also one of our main work. Because in the case of the same model accuracy, the simpler the model structure, the stronger the robustness. Therefore, when optimizing the network, we can reduce the amount of network parameters by adding new modules to ensure the network accuracy and good anti noise ability (reduce the sensitivity of the network to input errors).

Point 3: Also, in the encoder-decoder structure, the compression degradation affects system performance in a significant way. How this problem can be solved.

Response 3: Thank you for your question. Encoder decoder structure depends on the depth of the network to some extent. The higher the depth, the more complex the parameters, and better fitting can be achieved. However, after we add several structures to the model: Res2Netplus, feature fusion module, attention mechanism and dual branch decoder, we get better results compared with the same layer network. We also tried deeper, but deeper will cause over fitting. Compared with the very small accuracy improvement, it is not worth using compared with its disadvantages, such as larger network model, longer training time and worse robustness of the network. Therefore, we have mainly added various modules to overcome the system functions of compression degradation.

Point 4: The author said “a lightweight dual-branch encoder-decoder framework”. How they validate that the framework is lightweight?! I didn’t see any proof for this.

The authors should measure the computational efficiency in terms of execution time and time complexity notation.

Response 4: Thank you for your question and suggestion. This paper proposes a lightweight dual branch encoder decoder framework. The original manuscript submitted did not use many evaluation indicators, but only two indicators, one is the weight of the model and the other is the parameter of the model. In the comparison between the two, we find that our model is the lowest in comparison with the other five models. According to your suggestion, we have added a new time complexity indicator in table 1, that is, GFLOPs (Giga Floating Point of Operations), to measure the complexity of the network model.

Point 5: The procedure of setup the dataset for evolution is not clear. Please explain clearly how many images for getting the results.

Response 5: Thank you for your question. There are 137 image data sets, 10 image test sets and 4 image verification sets in the mass data set. We cut the data set first. The data processing strategy used in this experiment is to use sliding window clipping. On this dataset, we use 156 sliding windows to slide up and down, and crop horizontally and vertically. Then crop by sliding the window. The cut training set is packaged into 8768 512 * 512 images, and we manually screen out the images with errors in the real images. After selection, our training set contains 7488 256 * 256 images, 640 test sets and 256 verification sets, which constitute our training data set this time. It is described in Section 4.1.

Point 6: The dataset contains 151 high-resolution aerial 423 images. It is a small number for evaluation the model. Please use also another dataset with more number of images for evaluation. 

Response 6: Thank you for your suggestion. Massachusetts Buildings Dataset 1500× 1500 pixels per image, with an area of 2.25 square kilometers. Therefore, the entire data set covers about 340 square kilometers. The data are divided into 137 image training sets, 10 image test sets and 4 image validation sets. Maybe the test set accounts for a low proportion of the whole datasets, but the sample quality contained in it is higher than that of the training set, and there are a lot of building pixels in the picture. However, in order to increase persuasiveness, we adopted WHU Satellite dataset, which is collected from cities and remote sensing resources around the world, including QuickBird, worldview series, IKONOS, ZY-3, etc. It contains a total of 204 images with resolutions ranging from 0.3m to 2.5m. Because of the relationship between epidemic situation and time, we can't call too many server resources at home, so we use this data set. However, when we make the data set, we use the ratio of 3:1 between the training set and the test set. Because the data set contains a wide range of samples, it is very suitable to test the robustness of building algorithm. This work is not particularly perfect because of time. We will gradually improve the relevant work on this dataset in subsequent modifications.

Point 7: How the authors selected the important parameters of the model and their suitable values. Please explain that.

Response 7: Thank you for your question. When selecting model parameters, we use the method that each network model trains 100 epochs. At the end of the training, the first few lower ones are selected according to the loss of the training set and the loss of the validation set. Then load several weights into the network, use the test set to test the quantitative indicators and visual images of these weights, and select a weight with the best quantitative indicator and the clearest visual picture as the weight of the final network.

Round 2

Reviewer 1 Report

The authors have completed all of my previous comments. I suggest the work is enough to be published in the Remote Sensing journal. 

Author Response

Thanks for your approval.

Reviewer 2 Report

I think that the paper can be accepted in the current form. The quality of this paper has been much improved. 

Author Response

Thanks for your approval.

Reviewer 4 Report

There are still a number of comments should be addressed. They are as follows:

Point 1: Why you still capitalize the first character of the normal words like “Dataset” in “Massachusetts Buildings Dataset” in the whole manuscript.

Point 3: Also, in the encoder-decoder structure, the compression degradation affects system performance in a significant way. How this problem can be solved.
In the response of the authors, they said "Therefore, we have mainly added various modules to overcome the system functions of compression degradation." What are these modules? Please explain them.

Point 4: The authors should measure the computational time of testing and training for each model compared to their model.

Point 6: Please use another dataset with large number of images for testing and evaluation. 

Point 7: How the authors selected the important parameters of the model and their suitable values. Please explain that. 
I mean that you should use one of parameters selection methods to select the best values of model's parameters.

Author Response

Point 1: Why you still capitalize the first character of the normal words like “Dataset” in “Massachusetts Buildings Dataset” in the whole manuscript.

Thank you very much for your advice. The first letter of "dataset" in "Massachusetts Buildings Dataset" is capitalized since it is a complete term to name the dataset. Besides, another reviewer asked us to capitalize the ‘D’, which makes us confused to follow. We therefore searched other relevant papers. For example, ref [51]: “BRRNet: a full revolutionary neural network for Automatic building extraction from high resolution remote sensing images” cited the dataset as “Massachusetts Buildings Dataset”. To our understanding, both ways work, but keeping consistence is vital.

Point 3: Also, in the encoder-decoder structure, the compression degradation affects system performance in a significant way. How this problem can be solved.

In the response of the authors, they said "Therefore, we have mainly added various modules to overcome the system functions of compression degradation." What are these modules? Please explain them.

Thank you very much for your question. We have added following modules to overcome the system function of compression degradation.

(1) Res2Netplus: to increase the receptive field through hole convolution and its own features. The features extracted by three layers can achieve the effect of four and five layers of the original network.

(2) CBAM: to aggregate the features extracted from the backbone network. By adding this, the effect of shallow network can also achieve the effect of medium and deep network.

(3) Dual branch decoder: to make use of the two branches in order to complement each other's details in local and global, and therefore realize the effect of the previous deep decoder when restoring the features.

We have added the above contents into the corresponding position of the manuscript.

Point 4: The authors should measure the computational time of testing and training for each model compared to their model。

Thank you very much for your suggestion. We have added a new column for training time in table 1. And the training and testing times of each model are analyzed with other models in the table analysis.

Point 6: Please use another dataset with large number of images for testing and evaluation。

Thank you very much for your suggestion. We have added a multi-feature dataset called WHU Satellite dataset, formed by the fusion of multiple satellite datasets. This dataset is collected from cities and remote sensing resources around the world, including QuickBird, worldview series, IKONOS, ZY-3, etc. Some large datasets have a single building because of geographical restrictions. The WHU Satellite dataset, on the other hand, contains a far greater diversity of buildings than the average dataset. After data expansion, the quality and quantity of this dataset is very high, which is challenging for any model.

Your Point 6 comment also remind us to optimize the proportion of dataset, i.e. the way we divide the training, test, and verification images. So far, the division ratio of the training and testing parts of our new WHU Satellite dataset reaches 3:1.

Point 7: How the authors selected the important parameters of the model and their suitable values. Please explain that.

I mean that you should use one of parameters selection methods to select the best values of model's parameters.

Thank you very much for your question and suggestion.

In the experimental study, Sklearn's RandomizedSearchCV and GridSearchCV have been used to find the best parameters of the model, and to improve the prediction results and the confusion matrix.

The RandomizedSearchCV is a combination of parameters randomly selected from the hyper parameter space, and the parameters are selected by a given number of iterations. The GridSearchCV is a basic super parameter tuning technology, similar to manual tuning. It builds a model for each arrangement of all given super parameter values specified in the grid, evaluates and selects the best model.

Considering the large number of super parameters in our model, in order to ensure the balance between the optimal combination of parameters and calculation time, we first use random search to find the potential combination of super parameters, and then use grid search to select the optimal feature in the potential combination of super parameters.
